# Immunonutrition and SARS-CoV-2 Infection in Children with Obesity

**DOI:** 10.3390/nu14091701

**Published:** 2022-04-20

**Authors:** Enza D’Auria, Valeria Calcaterra, Elvira Verduci, Michele Ghezzi, Rossella Lamberti, Sara Vizzuso, Paola Baldassarre, Erica Pendezza, Veronica Perico, Alessandra Bosetti, Gian Vincenzo Zuccotti

**Affiliations:** 1Pediatric Department, “Vittore Buzzi” Children’s Hospital, 20154 Milan, Italy; enza.dauria@unimi.it (E.D.); valeria.calcaterra@unipv.it (V.C.); michele.ghezzi@asst-fbf-sacco.it (M.G.); rossella.lamberti@unimi.it (R.L.); sara.vizzuso@asst-fbf-sacco.it (S.V.); paola.baldassarre@unimi.it (P.B.); erica.pendezza@asst-fbf-sacco.it (E.P.); veronica.perico@studenti.unimi.it (V.P.); alessandra.bosetti@asst-fbf-sacco.it (A.B.); gianvincenzo.zuccotti@unimi.it (G.V.Z.); 2Pediatric and Adolescent Unit, Department of Internal Medicine, University of Pavia, 27100 Pavia, Italy; 3Department of Health Sciences, University of Milan, 20142 Milan, Italy; 4Department of Biomedical and Clinical Science “L. Sacco”, University of Milan, 20157 Milan, Italy

**Keywords:** SARS-CoV-2 infection, obesity, antiviral immunity, micronutrients, gut–lung axis, dysbiosis, immunonutrition

## Abstract

Since the beginning of the SARS-CoV-2 pandemic, there has been much discussion about the role of diet and antiviral immunity in the context of SARS-CoV-2 infection. Intake levels of vitamins D, C, B12, and iron have been demonstrated to be correlated with lower COVID-19 incidence and mortality. Obesity has been demonstrated to be an independent risk for the severity of COVID-19 infection in adults and also in children. This may be due to different mechanisms, mainly including the gut dysbiosis status observed in obese children. Moreover, the existence of a gut–lung axis added new knowledge to on the potential mechanisms by which diet and dietary substances may affect immune function. The aim of this narrative review is to address the intricate inter-relationship between COVID-19, immune function, and obesity-related inflammation and to describe the role of nutrients and dietary patterns in enhancing the immune system. Two ways to fight against COVID-19 disease exist: one with an antiviral response through immune system boosting and another with antioxidants with an anti-inflammatory effect. In the current pandemic situation, the intake of a varied and balanced diet, rich in micronutrients and bioactive compounds including fibers, should be recommended. However, clinical studies conducted on children affected by SARS-CoV-2 infection and comorbidity are warranted.

## 1. Introduction

As reported by World Health Organization (WHO) guidelines, it is crucial to follow a healthy lifestyle in order to guarantee optimal health status, which in turn supports immune cell systems and modulates systemic inflammation [1]. Nutrients have a potential to modulate, whether directly or indirectly, the development and function of innate and acquired immunity [2]. Immunonutrition is based on the concept that malnutrition impairs immune function and that specific nutrients may be used to modify inflammatory or immune responses [3,4].

Childhood obesity has become a serious public health problem in many parts of the world. Nearly 41 million children under 5 years of age and more than 340 million children and adolescents between the age of 5 and 19 were either overweight or affected by obesity as last reported [1]. Obesity is characterized by a state of low-grade chronic inflammation in addition to altered levels of circulating nutrients and metabolic hormones. In fact, adipose tissue is not only part of the endocrine system, but it is also an active immune organ suitable for managing the adaptive and innate immune responses playing an important role in adipose immunometabolism [5]. Obesity represents a risk factor for developing severe COVID-19 sequelae due to the high concentrations of cytokines produced in the meantime by adipose tissue and by innate immunity [6,7]. Additionally, gut micro-biota is a player in the maturation, development, and functioning of both the innate and adaptive immune system, as well as being a contributor to the development of the obese phenotype [8,9].

During SARS-CoV-2 infection in obese subjects, the organism fights both the acute inflammatory state triggered by the infection and additionally the low-grade chronic inflammation due to obesity. Thus, the immunonutrition may be important to support the immune response and at the same time to reduce inflammation caused by the virus and obesity. Additionally, a balanced diet is essential for weight control. Tailored immunonutrition in children with obesity should be an adjuvant treatment to reduce the risk of infections and the disease course in COVID-19 patients.

The aim of this review is to consider the relationship between SARS-CoV-2 infection, immune function, and obesity-related immune dysregulation and to describe the role of nutrients and dietary patterns in immunomodulation. A nutritional approach can be used to enhance the immune system’s response in children and adolescents with obesity in the COVID-19 era.

## 2. Methods

A narrative review was performed [10] with reference to the English literature literature available over the past 10 years. The most relevant published manuscripts, including original papers, metanalyses, clinical trials, and reviews were independently identified by the authors; case reports and case series were not considered. A search in PubMed, Scopus, EMBASE, and Web of Science was adopted. The following search terms (alone and/or in combination) were employed: COVID-19, SARS-CoV-2 infection, immunonutrition, obesity, adipose tissue inflammation, children, diet, nutrients, immunomodulation, and immune function. The contributions were critically reviewed and collected. The final version was approved by everyone.

## 3. Obesity and SARS-CoV-2 Infection in Children

### 3.1. Epidemiological Data

Children with COVID-19 present mild clinical characteristics and lower mortality rates compared to adults [11].

In children, SARS-CoV-2 infection generally occurs more frequently in a asymptomatic or mild form than in adults [12,13].

A Chinese large-scale pediatric study observed that 94.1% of children and adolescents affected by COVID-19 were asymptomatic, and only 6% of them developed severe symptoms, with a very low mortality (<1%) [14].

According to a Centers for Disease Control and Prevention (CDC) report, the commonest clinical conditions among patients hospitalized with COVID-19 were diabetes, chronic lung disease, and cardiovascular disease [15,16].

The relationship between obesity and viral diseases has been studied, showing impairment in immune response and inadequate vaccine response to influenza [17,18].

Obesity has only recently been identified as a risk factor for severe COVID-19 disease in children.

During the COVID-19 epidemic in Canada, obesity was the third most prevalent demographic factor among children admitted to intensive care units. In New York, obesity was the most prevalent comorbidity among severe cases of children with COVID-19 infection [19]. Obesity may predispose a patient to high severity and mortality due to COVID-19 even in young patients, through aspects related to obesity itself and also its comorbidities [20], although the exact pathogenetic mechanisms are far from being fully explained.

### 3.2. SARS-CoV-2 Infection in Children and Immune Function

The SARS-CoV-2 enters human cells through the metallo-carboxyl peptidase angiotensin receptor (ACE)-2 [21]. ACE-2 has an extracellular N-terminal domain with a catalytic site, a C-terminal membrane anchor, and a zinc-binding domain [22].

There are two forms of this receptor. One is located at the cell membrane level, allowing the link with the spike protein of SARS-CoV-2. The entry of SARS-CoV-2 into cells depends on the trans membrane serine protease 2 (TMPRSS2), which divides the spike protein into two subunits (S1 and S2 subunits), allowing viral fusion with cell membrane. The other is a soluble form without the trans membrane anchor and is found in the bloodstream, maintaining the capacity of binding to SARS-CoV-2 [23]. Increased level of soluble ACE-2 is associated with more severe disease, possibly due to an increase in angiotensin II (AngII) [24,25,26].

The binding of SARS-CoV-2 to the ACE-2 receptor causes its cellular internalization (down-regulation) and imbalance of both the renin–angiotensin–aldosterone system (RAAS) and the kinin–kallikrein system (KKS). Without the action of ACE-2, AngII increases vasoconstriction, inflammation, and oxidative stress [21].

Moreover, the KKS regulates several biological processes such as coagulation, inflammation, and pain. Through binding to bradykinin receptor-B2 (BRB2), bradykinin promotes the nitric oxide production that has a potent vasodilator effect, balancing the vasopressor effect of the RAAS. Bradykinin also regulates tissue plasminogen secretion (tPA), playing an important role in thrombus formation. ACE-2 also regulates the KKS [27,28,29].

ACE-2 down-regulation leads to an increased activation of NF-κB through the angiotensin receptor type 1 (AT1R) and mediated by protein kinases: mitogen-activated protein-kinase/extracellular signal axis-regulated-kinase (MAPK/ERK). This represents the mechanism of first-occurring NLR family pyrin domain containing 3 (NLRP3) inflammasome activation during SARS-CoV-2 infection [30,31].

NLRP3 inflammasome activation leads in turn to interleukin (IL)-1 β and IL-18 secretion. IL-1β plays a crucial role in the development of T helper 17 (Th17) through IL-1R signaling. Furthermore, the excessive activation of IL-1R signaling by IL-1β down-regulates transforming growth factor-β (TGF-β)-induced Foxp3 expression, with consequent further promotion of Th17 differentiation and concomitant suppression of naive (Cluster of Differentiation) CD4 differentiation in Threg [32].

Several studies have been performed to understand ACE-2 tissue expression. Surprisingly, the expression of ACE-2 is extremely low in the lung and localized in a small percentage of type II alveolar epithelial cells [33]. On the contrary, ACE-2 expression is higher in the small intestine, testis, kidney, heart muscle, colon, and thyroid gland [34].

Nasal epithelial cells are likely the primary site of SARS-CoV-2 infection mediated by ACE-2 expression, while infection of the lower respiratory tract may be due to aspiration. The SARS-CoV-2 infects type II pneumocytes and activates the lung-resident macrophages, causing local inflammation, increasing vascular permeability and attracting other inflammatory mediators. The accumulation of fluid in alveoli leads to dyspnea and pneumonia [35,36].

In SARS-CoV-2 infection, the release of interferons (IFNs) increases the expression of ACE-2 and contributes to tissue damage [37]. Other evidence [38] showed that IFNs actually induce the expression of a truncated isoform of ACE-2 which is unable to bind SARS-CoV-2 and does not potentiate its infection.

Pathogenetic mechanisms, other than the massive release of cytokines, include coagulopathy and endothelial dysfunction. The endothelial dysfunction enhances thrombin generation and inhibits fibrinolysis, leading to hypercoagulability [39]. Hypercoagulability itself is an important sign of inflammation due to IL-17, IL-6, and IL-8 [40,41].

In patients with poor clinical outcomes, an uncontrolled “cytokine storm” develops, with the production of pro-inflammatory mediators such as IL-6, tumor necrosis factor-α (TNF-α), and IL-1b.

The mortality and the morbidity related to COVID-19 are linked to an “immunological collapse”, with the loss of B and T cells in the spleen and secondary lymphoid organs: the inability to control viral replication in the early stages of disease compromises the clinical outcome [42].

Several studies have shown that the expression of ACE-2 is age-dependent and that the ACE-2 levels are lower in children than in adults [43]. Nevertheless, data on ACE-2 are contradictory: some authors found that the amount of ACE-2 protein in cells is lower in children than in adults [44] while others discovered the opposite [45,46]. However, the alternative angiotensin II receptor (ATR2) is more expressed in children than adults, leading to an anti-inflammatory role with the down-regulation of the proinflammatory receptor ATR1, competing with it for angiotensin II binding [43,47].

ACE-2 contributes essentially to reduce the inflammatory process with the down-regulating of the pro-inflammatory peptides of the renin–angiotensin–aldosterone system (RAAS) and the kinin–kallikrein system (KKS). The failure to revert the inflammation promoted by SARS-CoV-2 may explain the severity of infection itself [43,47]. ACE-2 receptors are up-regulated by type 1 inflammation (INF), suggesting that Th1/Th2 balance may help the development of SARS-CoV-2 infection. Th2-polarized cytokine production in children is presumably protective against SARS-CoV-2 infection [48,49].

Other theories can be considered in order to understand why SARS-CoV-2 infection in children is milder than in adults [50,51]. Firstly, the immune response is different. In children, the immune system is continuously stressed by natural exposure to viruses and vaccinations. For this reason, children produce SARS-CoV-2-neutralizing antibodies more effectively [50].

In addition, age-dependent defects in T and B cell function and the excess production of type 2 cytokines can lead to a deficiency in control of viral replication and more prolonged pro-inflammatory responses.

Moreover, the possibility of previous infections in children with human coronaviruses different from SARS-CoV-2 should confer partial protection against SARS-CoV-2 [52].

### 3.3. Obesity as a Risk Factor for Severe COVID-19 Infection in Children: What Is the Link?

Obesity represents one of the clinical conditions associated with the development of a more severe infection (worse outcome, more frequent complications, need of hospitalization, and critical illnesses) along with chronic respiratory and heart diseases, immunodeficiencies, neurological diseases, and metabolic diseases [53]. These pathological conditions are characterized by an altered release of cytokines, impairment of the immune response, and dysregulation of immune cell differentiation [54].

Obesity has been recognized as an independent risk factor for the severity of SARS-CoV-2 infection [55].

Unfortunately, during the COVID-19 pandemic, the change in eating habits due to the lockdown led to an increase in the rate of obesity (called “covibesity”) in children [56]. The prevalence of excess weight (overweight and obesity) increased from 23.9% in the pre-COVID-19 period to 31.4% in the COVID-19 period [57].

A recent systematic review and meta-analysis have demonstrated that for an obese child compared to a child without comorbidities, the relative risk of developing a severe form of COVID-19 is equal to 2.87, confirming the hypothesis that obesity represents an important risk factor for more severe disease linked to SARS-CoV-2 infection [53].

The exact pathogenetic mechanisms explaining the major risk of obese patients for more severe SARS-CoV-2 infection are not fully understood. The adipose tissue can increase susceptibility and progression to a more severe form of SARS-CoV-2 disease through several mechanisms, although the exact pathogenetic mechanisms are far from being fully understood. Various receptors required for SARS-CoV-2 infection are expressed in the adipose tissue. ACE2, the functional receptor for SARS-CoV and SARS-CoV-2, is highly expressed in adipose tissue, serving as a gateway to the virus and activating the IFN-alfa pathway [58].

The ACE2 hyper-expression in obesity could transform the adipose tissue into a potential viral target and reservoir [59].

In addition to ACE2, a potential SARS-CoV-2 receptor is dipeptidyl peptidase 4 (DPP4), a ubiquitous membrane-bound aminopeptidase involved in the regulation of glucose homeostasis and inflammation. Since it is up-regulated in obesity and especially in the insulin-resistance state, it should be considered another way of access to human cells for COVID-19, determining strong inflammation and immune response [60].

In other words, adipose tissue in obese individuals may act as a SARS-CoV-2 reservoir.

Furthermore, the low-grade systemic inflammation due to the excess of visceral fat represents a risk factor for more severe disease by activating different immune pathways.

Adipose tissue is composed of adipocytes surrounded by connective tissue containing fibroblasts, preadipocytes, and cells belonging to the native and adaptive immune system such as macrophages, mast cells, neutrophils, and T and B lymphocytes, making it an immune organ [61] (Figure 1).

The excessive fat mass and the hypertrophy of adipocytes determine hypoxia and intracellular oxidative stress [62,63], which in turn induce a chronic mild-grade systemic inflammation as shown by circulating levels of cytokines and acute phase proteins. This mild-grade inflammation plays a relevant role in the impairment of adipocyte function and immune state dysregulation of obese individuals compared to normal weight individuals [64].

In obese patients, hypertrophic adipose tissue determines macrophage activation and M1 polarization, associated with systemic inflammation and insulin resistance through TNF-alfa-inducible nitric oxide synthase (iNOS) pathways [64,65,66].

These inflammatory macrophages produce inflammatory cytokines such as IL-1β, IL-6, and IL-12. On the contrary, M2 macrophages, prevalent in lean individuals, produce anti-inflammatory cytokine-like IL-10 [67,68].

The second common immune cells present in adipose tissue are T lymphocytes (T cells, both CD4+ and CD8+ T) [69,70].

In particular, studies reported an increase in CD8+ T, CD4+ Th1, and Th17 compared with T reg and Th2 which have an anti-inflammatory function [65] in obese people compared to those with a normal weight.

CD8+ T cells and CD4+ Th1 secrete pro-inflammatory cytokines such as IFN-γ which seem to stimulate polarization M1 of the adipose tissue macrophages. Instead, the subtype CD4+ Th17 cells secrete IL-17 which is involved in the release of other inflammatory molecules such as IL-6, IL-21, IL-22, chemokines, metalloproteases (MMPs), and TNF-alfa [71].

By contrast, there is a decrease in T reg cells which normally suppress excessive inflammation processes through the secretion of anti-inflammatory cytokines such as TGF-β and IL-10 [72]. Thus, obesity leads to a disproportion between T reg and Th17, associated with the development and progression of insulin resistance and obesity-related complications [73,74].

Regarding B lymphocytes, they also secrete inflammatory cytokines, such as IL-2 and IL-12, which affect T cell polarization toward Th1 versus Th2 cells [64,75].

This varied population of lymphocytes contributes to perpetuate the inflammatory state in obesity.

Adipose tissue in obese people also hosts a large amount of sentinel cells belonging to the innate response (neutrophils, mast cells, and dendritic cells) which contribute to amplify the inflammatory process through secretion of pro-inflammatory mediators (TNF-alfa, IL-1β, IL-8, and macrophage inflammatory protein-1 alpha (MIP-1)) and promote the recruitment of different immune cells [69].

Conversely, the eosinophils and Th2 cytokines (IL4, IL10, IL13, and TGF-β) that are essential for M2 polarization of macrophages and anti-inflammatory activity are reduced [76], making obesity responsible for a down-regulation of anti-viral responses [16].

In addition, the adipose tissue secretes hormones such as leptin which are elevated in obese children. This hormone acts not only on signaling the hypothalamus to suppress appetite but also the immune system [77].

Leptin induces secretion of pro-inflammatory cytokine such as IL-6, TNF-alfa, and IL-1 by macrophages, promotes Th1 response, and suppresses T reg and Th2’s proliferation and activity.

In particular, IL-6 is secreted in the adipose tissue, promotes the polarization of macrophages to the pro-inflammatory phenotype, T cell survival, and resistance to apoptosis, and induces CD4+ T cell differentiation to the Th1 or Th17 subtypes [64,78].

Moreover, it directly stimulates secretion of C-reactive protein (PCR), an acute phase plasma protein which is elevated during all inflammatory processes.

On the contrary, adiponectin is an anti-inflammatory hormone with an insulin-sensitive action. Physiologically, adiponectin releases anti-inflammatory and immunosuppressive cytokines such as IL-10 and promotes polarization toward the M2 phenotype and differentiation in CD4+ Th2 subtype. It is secreted by healthy adipose tissue and its plasma levels are decreased in obesity [79,80].

Last but not least, IL-6 and TNF-α produced in adipose tissue are responsible for the gradual loss of function of adipose-derived mesenchymal stromal and stem cells (ASCs), reducing cell differentiation capacity, impairing ciliogenesis, and increasing the risk of lung injury through fibrosis during SARS-CoV-2 infection.

## 4. Nutrition and Immune Function

To date, the relevance of a status of well-being and a functioning immune system has come to be more noticeable. Many factors can contribute to the development of chronic illnesses causing a mild chronic inflammatory state [81], and diet plays a central role among environmental factors. It has been well demonstrated that the adoption of an unhealthy lifestyle is associated with the development of non-communicable diseases, probably mediated also by a high level of oxidative stress [82]. Furthermore, optimal nutrition can regulate immune maturation and response to inflammation. Requirements for nutrients may be raised in inflammatory conditions, mostly during infections, promoting a nutrient catabolic state [83]. Therefore, nutritional deficiencies may occur, establishing a vicious circle, leading to malnutrition associated with a weak immune function. Consequently, a balanced nutritional status is necessary to avoid and counter-infections [84].

Many dietary nutrients have an impact on immune-mediated diseases, but on the other hand, it is well recognized that immune system functions may depend on nutrient levels [3]. Further, nutrients play a role in physical barriers (gut mucosal barrier and skin), the immune system response (both innate and adaptive macrophage action and differentiation, B and T cell functions, etc.), and microbiome modulation [82]. At the same time, the immune system influences the requirement and metabolism of different nutrients.

The role of nutrition in the development of the immune response is necessary from intrauterine life. Studies conducted in animal models have shown that a micronutrient deficiency during pregnancy negatively affects the functional development and growth of the thymus and B lymphocytes. All of this increases the risk of infections during childhood and inflammatory illnesses at later ages [85]. Consequently, it is important to ensure a healthy diet from the prenatal era through to the first years of the child’s life, to ensure the development of an adequate immune system [86].

Breast milk, the first food to which newborns are exposed, is a key factor in immune system development. With its biologically active compounds (antimicrobial peptides such as defensins and cathelicidin), breast milk sustains active and passive immunity during the primary years of life [87].

Among the macronutrients, carbohydrates guarantee a healthy immune response of all the cells of the immune system that require glucose to meet their elevated metabolic requirements [88]. Especially galactose plays an important role in host defenses [89]. Proteins are involved in cytokine and lymphocyte production and gene expression and regulate immune cell activation (B and T lymphocytes, macrophages, and natural killers) [89]. A bioactive peptide contained in meat and fish called carnosine manages to regulate the immune system by raising the production of interleukin-1 and inhibiting the apoptosis of neutrophils [90]. Among fats, long-chain polyunsaturated fatty acids (LC-PUFAs) are essential for the proper functioning of the immune system because they ensure the cell membrane fluidity, modulate gene expression, and signal transduction and produce molecules with pro- or anti-inflammatory actions [91].

Additionally, micronutrients are essential for the proper functioning of the child’s immune system. Effectively, these regulate both innate and adaptive immunity, antibody reaction, cytokine activity, and the response of Th1 and Th2 lymphocytes [92,93]. Many vitamin trace elements (VTEs) such as iron, zinc, selenium, copper, chromium, vitamins A, C, D, E, and folic acid are critical in the maintenance of immune competence [92].

Therefore, the relationship between nutrition and immune function is intricate [82,94]. An unbalanced diet might affect the proper functioning of the immune system; this may be due to either inadequate intake of energy and macronutrients (carbohydrates, proteins, and fats) or micronutrient insufficiency (iron, selenium, zinc, and vitamins). No single food contains all the beneficial nutrients involved in immune function improvement. A varied and balanced diet can include all the components involved in these functions, acting in a synergistic way.

## 5. Dietary Patterns and COVID-19

Several studies have recently investigated whether nutrition can play a part in prevention of infection and if it can improve the outcome of the disease in patients affected by COVID-19.

It is well described in the literature that adopting healthy dietary patterns helps all cells, including those of the immune system, to function optimally [95]. Several studies have shown that nutritional status highly affects the immune system responses [95]: states of malnutrition, being overweight, and obesity negatively influence the immune system, leading to a higher risk of viral infections. Some reports [96] have demonstrated that a high body mass index (BMI) or extreme adiposity may be risk factors for complications during COVID-19 infection, not only in adults but also in children [53], probably because of the presence of various pulmonary diseases in overweight and obese people compared to normal weight subjects [97]. Moreover, patients with obesity, if affected by comorbidities (e.g., diabetes, arterial hypertension) that can impair their cardio-pulmonary function, may be at greater risk of developing severe COVID-19 disease [98]. It is therefore essential to maintain an adequate nutritional status and body composition according to the international recommendations, especially in this pandemic period [99]. It should be noted that obese children have increased cardiovascular and metabolic risk both in childhood and adulthood and may present with early signs of metabolic syndrome [100]. During the SARS-CoV-2 infection, there is an increased energy demand and thus an enhancement of the basal metabolic rate related to an activated immune system. For this reason, it is necessary to adopt a correct nutritional approach so as to obtain an excellent reaction of the immune cells in terms of their response to pathogens and to improve their reactivity when necessary [101]. However, some studies have pointed out that it is important to remember that improving nutritional status should not be judged satisfactory in the treatment of SARS-CoV-2 infection and COVID-19 adverse effects when it is adopted without other treatments [102]. An inadequate nutritional status seems to be a predictor of mortality in acute SARS-CoV-2 infection and critical illness, particularly for elderly or polymorbidity subjects [103].

Over the past year, many research groups have conducted studies to identify which nutritional strategies, acting as immunostimulators (with the modification of signaling molecules, influencing the cellular activation and gene expression) could improve the immune system response for the prevention, management, and recovery of COVID-19 patients.

Overall, there are two ways to fight against COVID-19: an antiviral response through immune system boosting and the use of antioxidants with anti-inflammatory effect.

Some authors have suggested including habitual diet foods with antioxidant function, such as vegetables, fresh fruits, soy, nuts [104], and omega-3 fatty acids (ω3) [105].

In a different study, the authors suggested following a diet based mainly on fresh food such as fruit, vegetables, whole grains, low-fat dairy, and good fats (olive oil and fish oil), reducing the intake of high-salt and high-caloric foods and sugary drinks during the COVID-19 pandemic [106].

## 6. Nutritional Supplements and COVID-19

The role of micronutrients in helping the immune system has been extensively investigated [102,107,108,109].

Several vitamins and trace elements take part in supporting the immune response and the lack of many of these micronutrients may damage several aspects of immunity (innate and acquired), increasing susceptibility to infections [110]. In fact, a recent review showed how viral infections can be caused by a compromised immune system, resulting in an insufficient reserve of micronutrients (both vitamins and trace elements) [111]. The intake of several substances (e.g., fatty acids, linoleic acids, essential amino acids, and vitamins and minerals), as shown in the literature, may improve the immune response, specifically when the immune system, as often happens in viral infections, can be influenced by their deficiencies [112].

Since the beginning of the SARS-CoV-2 pandemic, there has been much discussion about the role of different micronutrients and antiviral immunity in the context of SARS-CoV-2 and COVID-19 infection.

In this regard, the intake levels of vitamins D, C, B12, and iron have been demonstrated to be correlated with lower COVID-19 incidence and mortality. This effect is more evident in populations with lower micronutrient status [113].

Some recent reviews have examined how satisfactory nutritional intake, matched with the integration of various functional foods or bioactive compounds, aids to keep optimal levels in the human body by improving different aspects of immunity [101,108,114,115].

In Table 1, the possible immunomodulatory role of some bioactive compounds against COVID-19 are summarized.

To date, clinical trials studying bioactive dietary compounds mainly have involved adults. Some have included children, but clinical studies specifically conducted on obese children affected by SARS-CoV-2 infection are lacking.

Despite SARS-CoV-2 infection having a recent onset and nutrition giving the impression that it can condition the outcome of patients affected by COVID-19, no specific recommendations or guidelines for nutrition or nutritional supplementation in patients affected by COVID-19, particularly in obese children and adolescents affected by COVID-19, can be drawn [146,147,148,149,150].

## 7. The Gut–Lung Axis, Dysbiosis, and Dietary Immunomodulation

There are several studies showing that the intestinal and respiratory microbiota develop together after birth [151,152,153,154]. A number of bacteria that had appeared before in the gut are present in the airways, leading to the assumption that micro-aspiration may be one of the mechanisms in the development of airway microbiotas. Other factors involved are host elimination abilities (cough and mucociliary clearance), the immune system, and physical conditions such as pH or oxygenation.

The intestinal microbiota requires a specific microenvironment to maintain immune homeostasis and provide local and systemic signaling to the innate and adaptive immune system development [155]. Alterations of the intestinal microbiota play a negative role not only on the gut but also on the brain, liver, and lungs, reducing the efficacy of immune responses [156,157,158]. The reciprocal relationship between intestinal microbiotas and lungs defined as the “gut–lung axis” (GLA) is indispensable for the regulation of the immune functions in the respiratory tract (Figure 2) [159,160,161,162].

The extensive relationship between the intestinal and the respiratory tract depends both on bacterial product secretion into systemic circulation and the activation of immune cells localized in gut and lung lymph nodes.

Via the mesenteric lymph nodes, bacterial proteins and cellular fragments reach systemic and pulmonary circulation, stimulating immune cell (dendritic cells, macrophages, T and B lymphocytes, neutrophils, and plasma cells) activation.

The bacterial products are transferred to the mesenteric lymph node (MLN) by macrophages and dendritic cells (DCs), where they may prime naive B and T cells. Plasma cells, derived from differentiation of B cells, can cut down the production of anti-inflammatory molecules such as IL-10, leading to the priming and differentiation of T cells. T cells may leave the gut-associated lymphatic tissue (GALT) and reach the airways, where they can influence the pulmonary immune response [163].

Intestinal tract dysbiosis can induce lung inflammation through the GLA, increasing the disease susceptibility to and the morbidity of several clinical conditions such as allergies, asthma, pulmonary infections, cystic fibrosis, and others [154,158,164].

Although several studies have demonstrated that the primary direction of cross-talk occurs from the gut to the lung, communication in the opposite direction may occur. The patients affected by chronic lung disorders (asthma, chronic obstructive pulmonary disease, and cystic fibrosis) present not only with dysbiotic airway microbiotas but also gastrointestinal involvement such as irritable bowel syndrome [154,155,165,166].

The microbiota of the gastrointestinal and the respiratory tract has a dynamic composition that undergoes variations from the early stages of life based on the gestational period, the mode of delivery, and breastfeeding [167]. In addition, several exogenous factors such as diet, ‘hygiene hypothesis’, smoking, antibiotics, and probiotic bacteria contribute to the composition of the gut microbiota, decreasing beneficial bacterial species (Lactobacilli and Lactococci) by outgrowth of pathogenic species (Enterobacteriaceae) [168].

Ley et al. introduced the concept that dysbiosis is related to obesity and metabolic syndrome [169]. In obese children, a higher Firmicutes to Bacteroidetes ratio is described as compared to normal weight children [170,171]. The resulting dysbiosis alters the intestinal barrier, allowing the permeability to structural components of bacteria and activating inflammatory patterns that may be involved in the development of insulin resistance and the production of inflammatory cytokines [172]. Furthermore, other evidence suggests that the development of obesity is associated with complex interactions between genetic background, gut microbiota, and diet [173,174,175].

In any case, diet plays a pivotal role in shaping the gut microbiome, promoting the growth of different bacterial species. A high-fat and high-protein western diet influences intestinal permeability, contributes to a pro-inflammatory microenvironment, and is closely associated with airway hyperresponsiveness [176,177,178]. On the contrary, a consistent fermentable fiber intake leads to production of short-chain fatty acids (SCFAs), known for their protective role against inflammation [165,179,180,181].

Furthermore, SCFAs maintain a balanced host–microbe relationship in the gut, eliciting mucus and antimicrobial peptide production, immunoglobulin A secretion, and enhancing intestinal epithelial barrier function, fortifying tight junction permeability [154,178,182]. SCFAs can also induce NLRP3 inflammasome activation [183]. Similar to what happens in the intestinal tract, SCFAs are able to promote T cell differentiation into Th1 and Th17 effector cells and IL- 10+Tregs [184].

As reported in a recent study, the microbial composition of the upper respiratory tract was affected by respiratory infections [185]. For instance, Prevotella and Flavobacterium species were predominant in case of Pseudomonas aeruginosa infection, while the Neisseria species was more represented in the Haemophilus influenzae infection [186]. Moreover, patients affected by influenza, parainfluenza, rhinovirus, coronavirus, adenovirus, or metapneumovirus infection seem to present a higher prevalence of Haemophilus and Moraxella colonization [154,161,164].

Regarding the SARS-CoV-2 infection, pneumonia and acute respiratory distress syndrome (ARDS) could impact the composition of the gut microbiota [187]. On the other hand, gut microbiota “dysbiosis” plays a key role in the pathogenesis of ARDS and sepsis [151].

Elderly and immunosuppressed patients present worse clinical conditions in case of SARS-CoV-2 infection; thus, it can be assumed that the clinical outcome depends on the cross-talk between the lung and the gut microbiota. Maintaining a healthy gut microbiome is essential to guarantee an optimal immune system status, preventing the damage to lungs and vital organ systems [144].

A balanced gut microbiota composition influences the effectiveness of lung immunity [188]. Some species of bacteria such as Lactobacillus Rhamnosus GG (LGG) and Bifidobacterium lactis play an anti-inflammatory role by reducing the number of Th2 and Th17 lymphocytes and increasing the expression of Treg lymphocytes [154,189,190]. Administration of certain bifidobacterial and lactobacilli has a beneficial impact on virus clearance from the respiratory tract, improves levels of INFs, and increases the innate and adaptive immune response. There is also evidence that probiotic administration modifies the dynamic balance between pro-inflammatory and anti-inflammatory mediators, reducing immune response-mediated damage to the lungs and preventing ARDS in SARS-CoV-2 infection [143].

Dysbiosis of the gut and airway microbiota is linked to chronic inflammatory disorders and infections. In recent years, it has been understood that the gut microbiota can play a critical role in influencing immune responses of other organs, including the lung. The existence of the GLA brings out new possibilities for therapeutic approaches to inflammatory diseases, involving the microbial intestinal and pulmonary microenvironment.

## 8. Conclusions

The relationship between gut, nutrition, and defense from infection is of relevance for maintaining a healthy status. Both the maintenance of health and the prevention of infectious disease in the respiratory tract are critically dependent on a proper functioning of the immune system.

When the immune system is dysregulated, as occurs in some chronic conditions such as obesity, the risk of more severe infections increases.

In the current situation of the COVID-19 pandemic, the role of diet as an immunomodulatory factor may be useful to help the immune system fight against the infection, especially in at-risk individuals such as obese children.

Thus, the intake of a varied and balanced diet rich in micronutrients and bioactive compounds including iron, selenium, zinc, vitamins, fibers, and ω3 fatty acids should be recommended.

As shown by several studies, a diet rich in fresh food (such as fruit, vegetables, whole grains, low-fat dairy, olive oil, and fish oil) and low in high-salt and high-caloric foods and sugary drinks, according to Mediterranean Diet principles, can reduce infection and mortality from COVID-19.

In some cases, dietary intake alone is not sufficient and specific supplementations are required.

For instance, several studies have highlighted that a daily intake of 400–1200 IU of VD for 6–12 months in children, increased in obese children and adolescents more at risk of hypovitaminosis, can prevent COVID-19 infection.

The literature is showing promising results for the supplementation of probiotics and DHA. However, to date, the current evidence does not allow us to draw specific indications regarding dosages in children, especially those with obesity.

Clinical studies investigating bioactive dietary compounds in children affected by SARS-CoV-2 infection and comorbidity are warranted.

## Figures and Tables

**Figure 1 nutrients-14-01701-f001:**
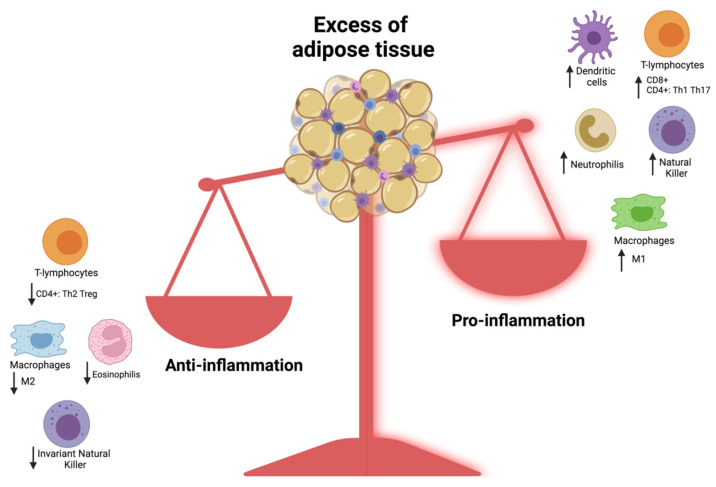
Pro-inflammatory and anti-inflammatory immune effects of excessive adipose tissue.

**Figure 2 nutrients-14-01701-f002:**
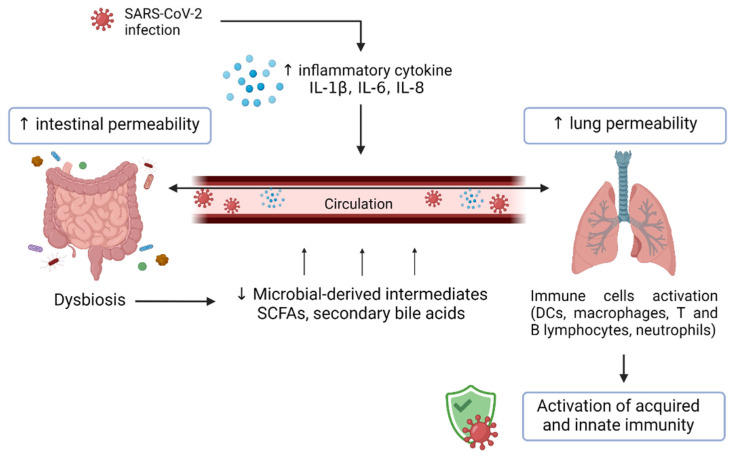
The gut–lung axis in SARS-CoV-2 infection. SCFA = short chain fatty acids; DCs = dendritic cells.

**Table 1 nutrients-14-01701-t001:** Roles of some bioactive compounds against COVID-19.

Nutrient/Bioactive Compound	Positive Suggested Role against COVID-19	References
Vitamin D	- Young children, elderly, and obese people are most at risk of hypovitaminosis D (caused by an insufficient sun exposure or by a diet low of VD-rich food). - In children, VD insufficiency (25-OH VD < 30 ng/mL) or deficiency (25-OH VD < 20 ng/mL) gives higher risk for respiratory infections. - VD supplementation lowers COVID-19 severity in hospitalized patients. - Promotes the production of antimicrobial molecules, activates defensive cells to destroy the virus, and decreases in vivo the production of inflammatory cytokines, preventing the cytokine storm. - Crucial regulator of the renin-angiotensin system (angiotensin converting enzyme 2, ACE2), useful for SARS-CoV-2 to move into the host cells. - Its supplementation can inhibit the transmission of the infection and avoid progression to severe disease. - In most studies in children, the intervention to prevent COVID-19 (daily intake 400–1200 IU for 6–12 months, considering higher doses in patients with VD deficiency or insufficiency, i.e., obese children and adolescents).	- Laird E et al., 2020 [116]. - Pecora F et al., 2020 [117]. - Entrenas Castillo M et al., 2020 [118]. - Giannini S et al., 2021 [119]. -Kumar R et al., 2021 [120]. - Costagliola G et al., 2021 [121]. - Martineau AR et al., 2017 [122].
Vitamin A	- Regulates both innate immune response (through natural killer cells, macrophages, and neutrophils) and adaptive immunity - During the initial phase of SARS-CoV-2 infection, the innate immune system acts on it by releasing IFN-1. Retinoids (in particular retinoid acid), for its immune-modulating properties, may improve IFN-1 activities. - Vitamin A (VA) and retinoid could be tested as antiviral substances in preclinical trials for COVID-19 treatment. To date, limited human clinical trials are ongoing (IRCT20180520039738N2, IRCT20170117032004N3), but there is no direct evidence of the efficacy of VA supplementation in patients affected by COVID-19.	- Yu-Ju La et al., 2021 [123]. - Trasino SE., 2020 [124]. - Jee J et al., 2013 [125]. - West CE et al., 1992 [126].
Vitamin C	- Influences functioning of the immune system (growth and function of both innate and adaptive immune cells, phagocytosis and microbial killing, antibody production, and generation of reactive oxygen species (ROS) and supportive epithelial barrier integrity). - VC deficiency is linked to an increased predisposition to severe respiratory infections such as pneumonia both in children and adults. - Ascorbic acid may inhibit the expression of ACE2 in human small alveolar epithelial cells, limiting the entrance of SARS-CoV-2.	- Maggini S et al., 2007 [92]. -Verduci E et al., 2021 [84]. - Pecora F et al., 2020 [117]. - Ivanov V et al., 2021 [127].
Vitamin E	- Antioxidant role (lowering the production of superoxides). -Supports T cell-mediated functions, optimization of Th1 response, and suppression of Th2 response.	- Yu-Ju Laia et al., 2021 [123]. - Gasmi A et al., 2020 [128].
PUFAs and DHA	- DHA has anti-inflammatory and antioxidant properties when enzymatically converted to specialized pro-resolving mediators (SPMs) known as resolvins, protectins, and maresins and increases immune system activity by helping to resolve the inflammatory response. - In an observational study conducted in children affected by MIS-C (multisystemic inflammatory syndrome), COVID-19-related, evidence of fatty acid (FA) alterations has been shown, suggesting a significant contribution of ω-6 FAs (linoleic acid and arachidonic acid) to the observed inflammatory state and supporting a possible dietary intervention to re-establish an appropriate balance among the FAs capable of promoting the resolution of the observed inflammatory condition.	- Verduci E et al., 2021 [84]. - Verduci E et al., 2021 [129].
Zinc	- Anti-inflammatory and antioxidant, reduces ROS in viral infections. - Direct role in antiviral activity by inhibiting viral replication on different pathogens, including SARS-CoV-2 and RSV, through the interference with the function of RNA-dependent RNA polymerase. - Promotes the proliferation and differentiation of T cells. - Activates the transcription factor FOXP3, implicated in the differentiation of Tregs (which produces anti-inflammatory cytokines such as IL-10), regulates the Th1/Th2 balance, the participates in the proliferation of Th17 cells.	- Calder PC et al., 2020 [102]. - Shakoor H et al., 2021 [130]. - Wessels I et al., 2020 [131]. - Razzaque MS. 2020 [132]. - Pal A et al., 2020 [133]. - Costagliola G et al., 2021 [121].
Lactoferrin	- Antiviral effect with the impairment of viral anchoring on the cellular surface by preventing the interaction between the virus and heparin sulfate glycosaminoglycan, and the inhibition of viral replication. Facilitates the clearance of the infectious agent, enhancing the activity of macrophages, neutrophils, and natural killer cells. - Eases the antigen presentation to T cells and modulates the secretion of pro-inflammatory cytokines (IL-6) which has a pivotal role in the pathogenesis of ARDS and MIS-C in pediatric patients with COVID-19. Decreases the expression of different chemotactic factors and adhesion molecules. - Reduces the damage derived from the production of ROSs. - May represent one of the factors contributing to the lower incidence and severity of COVID-19, especially by decreasing incidence of clinically relevant disease in the newborn.	- Chang R et al., 2020 [134]. - Lang J et al., 2011 [135]. - Peroni DG, Fanos V. 2020 [136]. - Peroni DG. 2020 [137]. - Kruzel ML et al., 2017 [138]. -Rosa L et al., 2017 [139]. - Siqueiros-Cendón T et al., 2014 [140].
Selenium	- Antioxidant role, ROS balance in inflammatory processes, immune cell function.	- Calder PC et al., 2020 [102]. - Shakoor H et al., 2021 [130]. - Bae M, Kim H. 2020 [141]. - Zhang J et al., 2020 [142].
Probiotics	- Influence both the transmission of SARS-CoV-2 and the immune balance of the host. - Particularly in children, probiotics may interfere with this mechanism, reinforcing the gut epithelial barrier and directly competing with the proliferation of SARS-CoV-2. - Specific probiotics may enhance local and systemic immune response, creating a “gut–lung axis” which finally favors the clearance of the infectious agent. - Influence the gut microbiome increasing local and systemic production of different proinflammatory cytokines with antiviral activity. - Enhance the activity of innate and adaptive immune system including the increased function of toll-like receptors (TLR) and the influence on the function of antigen-presenting cells. - May moderate the systemic levels of pro-inflammatory cytokines associated with the “cytokine storm”, and cause a rise of serum anti-inflammatory cytokines such as IL-10.	- Baud D et al., 2020 [143]. - Dhar D, Mohanty A., 2020 [144]. - Lei WT et al., 2017 [145]. - Costagliola G et al., 2021 [121].

VD = vitamin D; IU = international units; IFN 1 = interferon 1; VA = vitamin A; VC = vitamin C; RSV = Respiratory syncytial virus; FOXP 3 = forkhead box P3; ARDS = acute respiratory distress syndrome.

## Data Availability

Not applicable.

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
