# Peer review of "Immunonutrition and SARS-CoV-2 Infection in Children with Obesity"

_nutrients, 2022, doi:10.3390/nu14091701_

Round 1
Reviewer 1 Report
Very nice review!
For my better understanding of the section "3. Covid 19 and immune function" a graphical illustration would help me to digest the massive amount of information.
Author Response
Dear Editor,
Thanks for the opportunity to review the manuscript. We would like to thank the Reviewers for taking time to review the manuscript. We believe that their valuable comments let us to greatly improve the manuscript. Point-by point responses, as following:
Reviewer 1:
For my better understanding of the section "3. Covid 19 and immune function" a graphical illustration would help me to digest the massive amount of information.
R: Thanks for your suggestion. A graphical abstract has been added to clarify the section 3
Reviewer 2 Report
Review of “Immunonutrition and SARS-CoV-2 infection in children with obesity” by Enza D'Auria et al.
The topic of this review article may be interested by many and has a wide impact potentially. The authors have collected abundant information regarding SARS-CoV-2 infection, obesity, and nutritional supplement which is very impressive to me. However, it is not clear that how putting this information together could improve our understanding of SARS-CoV-2 infection in children with obesity. This review may need a bit of work. The authors may consider improve the review by these aspects:
1. Try to link information together instead of listing them. For instance, in page 3 authors discussed the immune response for Covid 19 infection, and in page 6 discussed what are currently known of inflammation in obesity. Can this information lead to a hypothesis of why obesity Covid 19 patients have a higher risk? If so, what is the major immune response that cause the increase of the risk? Any study supports this?
2. Guide readers to your conclusion. For example, line 212-217, “Evidences from adults show that in obese asthmatic patients prevails a Th1/Th17 in flammation pattern in the airways, with significant neutrophilia and elevated levels of TNF-α, IFN-γ and IL-6[51]. On the contrary, in childhood asthma–obesity phenotype, the classic atopic Th2 pattern predominates, with eosinophilic inflammatory infiltrate and high levels of IL-4, IL-5 and IL-13, which could hypothetically be a protective factor for severe SARS-CoV-2 infection in children [52]”. It is not clear that why high levels of IL-4, IL-5 and IL-13 could protect?
3. Provide more details of the known studies. Is this study done in mice or human patient, for which disease/infection? Does the finding depend on any special condition? Is the conclusion generally agreed or is there conflict evidence? Information like this would help readers to understand better.
Author Response
Dear Editor,
Thanks for the opportunity to review the manuscript. We would like to thank the Reviewers for taking time to review the manuscript. We believe that their valuable comments let us to greatly improve the manuscript. Point-by point responses, as following:
Reviewer 2:
The topic of this review article may be interested by many and has a wide impact potentially. The authors have collected abundant information regarding SARS-CoV-2 infection, obesity, and nutritional supplement which is very impressive to me. However, it is not clear that how putting this information together could improve our understanding of SARS-CoV-2 infection in children with obesity. This review may need a bit of work. The authors may consider improve the review by these aspects:
- Try to link information together instead of listing them. For instance, in page 3 authors discussed the immune response for Covid 19 infection, and in page 6 discussed what are currently known of inflammation in obesity. Can this information lead to a hypothesis of why obesity Covid 19 patients have a higher risk? If so, what is the major immune response that cause the increase of the risk? Any study supports this?
R: Thank you for your valuable comments. Accordingly, we completely edited the paragraphs 3 and 4 as following: paragraph 3 has been divided in three different sections to make clearer to the readers; in the section 3 c we explain the hypotheses of why obesity have a high risk of more severe infection from SARS CoV2. Two hypotheses are discussed: the hypothesis of adipose tissue as reservoir for the virus and the immune dysregulation due to the low-grade inflammatory state, characteristic of obesity. Recent publications have been added to support the hypotheses of why obesity Covid 19 patients have a higher risk.
- Guide readers to your conclusion. For example, line 212-217, “Evidences from adults show that in obese asthmatic patients prevails a Th1/Th17 inflammation pattern in the airways, with significant neutrophilia and elevated levels of TNF-α, IFN-γ and IL-6[51]. On the contrary, in childhood asthma–obesity phenotype, the classic atopic Th2 pattern predominates, with eosinophilic inflammatory infiltrate and high levels of IL-4, IL-5 and IL-13, which could hypothetically be a protective factor for severe SARS-CoV-2 infection in children [52]”. It is not clear that why high levels of IL-4, IL-5 and IL-13 could protect?
- Line 212-217 have been deleted as we believe that they are not relevant to the issue.
- Provide more details of the known studies. Is this study done in mice or human patient, for which disease/infection? Does the finding depend on any special condition? Is the conclusion generally agreed or is there conflict evidence? Information like this would help readers to understand better
- R. Thank you for your suggestion. Editing the main paragraphs about the link between obesity and SARS-CoV-2 infection in children, we have considered the latest evidences from human studies about the low-grade inflammation related to the excess of adipose tissue and the probably pathogenetic mechanisms associated to more severe viral infection.
Reviewer 3 Report
The aim of this manuscript is to review the mechanism of COVID-19 pneumonia and the relationship between COVID-19 and children with obesity. The author reviewed the risk factors and mechanism of SARS-CoV-2 infection. The immune response of COVID-19 and the role of obesity and nutrition/dietary in immune system. They also reviewed the effect of nutritional supplements in against the COVID pneumonia and the theory of gut-lung axis in COVID-19 infection.
- Since the COVID pandemic, many reviews were published to discussed the role of obesity, metabolic syndrome and COVID pneumonia. The special insight of this manuscript should focus on the children’s immune system, obesity child and its role in Covid infection (The title is very interesting, but the content of the manuscript contain very less discussion about COVID in pediatric)
- The child immune system is different from adult, please discuss more in detail.
- A comparison between child and adult immune system should be reviewed and discussed.
- The author should provide the epidemiology of Covid infection in child, child with obesity and compare it with healthy adult and obese adult.
- The role of Nutrition/dietary in child immune systems should be discussed.
- The statement of Line 406-407 should be revised. COVID pneumonia treatment guideline includes Dexamethasone, JAK inhibitor and IL-6 inhibitor, which mainly target on anti-inflammation. Antioxidant have been showed to have some effect, but not as effective as above.
Author Response
Dear Editor,
Thanks for the opportunity to review the manuscript. We would like to thank the Reviewers for taking time to review the manuscript. We believe that their valuable comments let us to greatly improve the manuscript. Point-by point responses, as following:
Reviewer 3:
The aim of this manuscript is to review the mechanism of COVID-19 pneumonia and the relationship between COVID-19 and children with obesity. The author reviewed the risk factors and mechanism of SARS-CoV-2 infection. The immune response of COVID-19 and the role of obesity and nutrition/dietary in immune system. They also reviewed the effect of nutritional supplements in against the COVID pneumonia and the theory of gut-lung axis in COVID-19 infection.
- Since the COVID pandemic, many reviews were published to discussed the role of obesity, metabolic syndrome and COVID pneumonia. The special insight of this manuscript should focus on the children’s immune system, obesity child and its role in Covid infection (The title is very interesting, but the content of the manuscript contain very less discussion about COVID in pediatric)
- The child immune system is different from adult, please discuss more in detail.
- A comparison between child and adult immune system should be reviewed and discussed.
- The author should provide the epidemiology of Covid infection in child, child with obesity and compare it with healthy adult and obese adult.
- The role of Nutrition/dietary in child immune systems should be discussed.
- The statement of Line 406-407 should be revised. COVID pneumonia treatment guideline includes Dexamethasone, JAK inhibitor and IL-6 inhibitor, which mainly target on anti-inflammation. Antioxidant have been showed to have some effect, but not as effective as above.
R: Thank you for your suggestion. We added a bit on epidemiological data in Covid 19 infection focusing on children, giving the Title of the review; as well as, we focused on Covi 19 infection and immune system in children, underlying the main difference compared to adults (especially focusing on ACE2 receptors differences between children and adults and plausible hypotheses explaining less severe infection in children than adults. Furthermore, we discuss the role of specific immune dysregulation in obesity in pediatric age, as detailed in the paragraph 3 and 4, that are edited, pointing out to the differences between children and adults. The role of nutrition/dietary in child immune system has been described in paragraph 4, 5 and 6, focusing mostly on the role of micronutrients as immunomodulators during Covid 19 infection. COVID pneumonia treatment was no longer mentioned as we believe that it is not relevant to the issue.
Round 2
Reviewer 2 Report
The authors have done a great job and addressed my questions.